# Poly-L-Lysine to Fight Antibiotic Resistances of *Pseudomonas aeruginosa*

**DOI:** 10.3390/ijms24032851

**Published:** 2023-02-02

**Authors:** Adeline Cezard, Delphine Fouquenet, Virginie Vasseur, Katy Jeannot, Fabien Launay, Mustapha Si-Tahar, Virginie Hervé

**Affiliations:** 1INSERM, Centre d’Etude des Pathologies Respiratoires (CEPR), UMR 1100, 37000 Tours, France; 2Université de Tours, Faculté de Médecine, 37000 Tours, France; 3UMR 6249 Chrono-Environnement, UFR Sciences Médicales et Pharmaceutiques, Université de Bourgogne-Franche Comté, 25030 Besançon, France; 4French National Reference Centre for Antibiotic Resistance, 25030 Besançon, France; 5Département de Bactériologie, CHU de Besançon, 25030 Besançon, France

**Keywords:** antibiotic resistance, *Pseudomonas aeruginosa*, cationic peptide, antibiotic combination

## Abstract

*Pseudomonas aeruginosa* is a major hospital-associated pathogen that can cause severe infections, most notably in patients with cystic fibrosis (CF) or those hospitalized in intensive care units. Given its remarkable ability to resist antibiotics, *P. aeruginosa* eradication has grown more challenging. Therefore, there is an urgent need to discover and develop new strategies that can counteract *P. aeruginosa*-resistant strains. Here, we evaluated the efficacy of poly-L-lysine (pLK) in combination with commonly used antibiotics as an alternative treatment option against *P. aeruginosa*. First, we demonstrated by scanning electron microscopy that pLK alters the integrity of the surface membrane of *P. aeruginosa*. We also showed using a fluorometry test that this results in an enhanced permeability of the bacteria membrane. Based on these data, we further evaluated the effect of the combinations of pLK with imipenem, ceftazidime, or aztreonam using the broth microdilution method in vitro. We found synergies in terms of bactericidal effects against either sensitive or resistant *P. aeruginosa* strains, with a reduction in bacterial growth (up to 5-log_10_ compared to the control). Similarly, these synergistic and bactericidal effects were confirmed ex vivo using a 3D model of human primary bronchial epithelial cells maintained in an air–liquid interface. In conclusion, pLK could be an innovative antipseudomonal molecule, opening its application as an adjuvant antibiotherapy against drug-resistant *P. aeruginosa* strains.

## 1. Introduction

Current trends in the inappropriate and overuse of antibiotics have propelled antimicrobial resistance as a global public health issue, which is projected to cause 10 million deaths per year worldwide by 2050 [1]. Among the bacterial species with the greatest resistance potential, six highly virulent and antibiotic-resistant bacterial pathogens (*Enterococcus faecium*, *Staphylococcus aureus*, *Klebsiella pneumoniae*, *Acinetobacter baumannii*, *Pseudomonas aeruginosa*, and *Enterobacter* spp.) are referred to as the ESKAPE group [2]. *Pseudomonas aeruginosa* is one the highest priority pathogens for drug development because of its antibiotic resistance and extraordinary adaptability and persistence. This aerobic Gram-negative bacterium is one of the main causes of hospital-acquired infections in the world and is the most common pathogen associated with disease exacerbations in patients with chronic respiratory illnesses such as cystic fibrosis (CF) or chronic obstructive pulmonary disease (COPD) [3,4,5].

The only way of improving patient outcomes with chronic *P. aeruginosa* lung infection is by using an inhaled antibiotherapy [6,7]. Unfortunately, antibiotic pressure promotes the emergence of resistant strains of *P. aeruginosa* that can lead to infections that are even more difficult to eradicate, causing longer hospital lengths of stay, higher readmission rates, and higher mortality [8].

*P. aeruginosa* infections have become an extremely significant and lethal problem, resulting in a total of 559,000 deaths each year due to severe treatment choice restrictions brought on by antimicrobial resistance [9]. Despite significant efforts, no new antibiotic classes effective against Gram-negative infections have been approved in more than 50 years. Thus, it becomes urgent to find new therapeutic strategies to fight these resistant strains of *P. aeruginosa*.

*P. aeruginosa* is an opportunist pathogen with natural and acquired resistance to several antibiotics. Concerning its natural resistance, four principal mechanisms are known [10]. The first one concerns its external membrane through which only a few molecules can pass through. Thus, some aminosides, carbapenems, and other β-lactamins have limited passage. The second natural resistance is the synthesis of different β-lactamases which hydrolyze β-lactamins (cephalosporinase inducible AmpC or oxacillinase called OXA-50 or PoxB). The third resistance mechanism is the production of different active efflux systems such as (i) MexAB-OprM efflux pump (constitutive expression), which generates resistance to several β-lactamins, fluoroquinolones, tetracyclines, trimethoprim, and chloramphenicol, or (ii) MexXY/OprM efflux pump (inducible expression by antibiotics), which generates resistance to aminoglycosides, fluoroquinolones, zwitterionic cephalosporins, or tetracyclines [11,12,13]. All of these mechanisms contribute to inactivating antibiotics or preventing them from reaching their target. The fourth natural mechanism of resistance is the biofilm which possess a very particular architecture, with a gradient of nutrients and oxygen availability. Thus, the bacteria located in depth within the biofilm will have a reduced metabolic activity but will be protected from phagocytosis and less sensitive to antibiotics [14]. Moreover, this lifestyle may enhance antibiotic resistance through a horizontal gene transfer process [15].

Three principal mechanisms of acquired resistance are described: (1) the overproduction of β-lactamases (e.g., cephalosporinase AmpC); (2) the overproduction of efflux systems (e.g., MexAB-OprM), and (3) the alteration of the porin OprD preventing carbapenem passage.

As a result of the selection pressure brought on by the widespread use of antibiotics, many *P. aeruginosa* strains that cause chronic infections in CF patients have gained resistance. Moreover, an increased number of resistant and multiresistant strains have been reported during the last 20 years, resulting in multidrug resistance of 25% of clinical *P. aeruginosa* isolates worldwide [16,17]. A bacterial strain is qualified for multidrug resistance (MDR) when it can resist three or more antibiotics classes. The last treatment for these MDR strains is colistin, which destabilizes the external bacteria membrane. However, certain strains of *P. aeruginosa* also develop resistance to this latter antibiotic [18].

The research of new therapeutic strategies is thus essential to treat patients. Recent studies are related to the improvement of formulations, antibiotic associations, or routes of administration [19]. Other studies explore the modulation of the immune response to facilitate the elimination of *P. aeruginosa*, by either a vaccine approach [20,21], by potentiating human antimicrobial peptides [22], or by using bacteriophages.

Previously, we showed that polycationic peptides have properties of interest for CF patients. α-Poly-L-Lysine (pLK) liquefies CF sputum by DNA condensation and exhibits antibacterial activity against *P. aeruginosa* [23]. More recently, we revealed that pLK eliminates the *P. aeruginosa* biofilm and alters the bacterial membrane [24]. Based on these data, this study aimed to evaluate the effect of combining pLK with antibiotics to counter *P. aeruginosa* resistance mechanisms.

## 2. Results

### 2.1. Permeabilization of P. aeruginosa Membrane by pLK

*P. aeruginosa* morphology was examined by scanning electron microscopy to visualize the effect of pLK on bacteria (Figure 1A). The bacterial surface was uniform and smooth without pLK (Figure 1A, left panel); however, following incubation with 10 μM pLK, alterations occurred on the bacterial surface. (Figure 1A, right panel). This result may suggest permeabilization of the bacterial membrane.

To investigate the ability of pLK to depolarize the bacterial membrane, we used DiSC3(5), a membrane potential-dependent probe. Upon membrane permeabilization, the potential is dissipated, and DiSC3(5) is released into the medium leading to a consequent increase in fluorescence. The pLK was tested at three concentrations (1, 10, and 100 µM) in PBS for 90 min. As expected, negative control (PBS) showed no release of relative fluorescence (Figure 1B). The pLK tested at 1 μM induced a small effect, with an increase in relative fluorescence around 2.5-fold after 15, 30, 60, or 90 min (Figure 1B). The pLK at 10 µM induced a membrane depolarization increase over time and reached a 10-fold increase in fluorescence after a 90-min incubation. Results obtained with 100 µM pLK also showed a time-dependent increase in relative fluorescence with a maximum of around 12.5-fold after 90 min (Figure 1B). Altogether, these results indicated the ability of pLK to permeabilize the membrane of *P. aeruginosa*.

### 2.2. In Vitro Synergistic Effect of pLK and Imipenem Association against Reference (PAO1) and Imipenem-Sensitive P. aeruginosa Strains

We considered if the pLK permeabilization ability would improve antibiotic passage and increase its efficacy. Different combinations of pLK and imipenem were evaluated against a reference wild-type strain, PAO1, and clinical strains, which are imipenem-sensitive. First, concerning the PAO1 strain, minimal inhibitory concentrations (MIC) were determined. For pLK, the MIC was 5 µM (50 mg/L) and for imipenem, the MIC was 4 mg/L. Several concentrations of pLK and imipenem were tested, and the concentration of each component showing no bacteriostatic effect was retained. The results showed that the combined effect of pLK 2 µM (CMI/2.5) and imipenem 1 mg/L (CMI/2) is greater than the additive effect of each molecule individually (Figure 2A,B).

Moreover, a bactericidal effect was also demonstrated with this combination (Figure 2C). Indeed, we determined a reduction in bacterial growth of 7-log_10_ compared to the control (1 × 10^7^ CFU/mL at the beginning of the experiment). Almost the same results were obtained against an imipenem-sensitive clinical strain, highlighting a synergistic effect with the association of 1 µM pLK (CMI/5) and 1 µg/mL imipenem (CMI/4) (Figure 2D). We also observed a bactericidal effect, resulting in a reduction in bacterial growth of 4-log_10_ compared to the control (1 × 10^5^ CFU/mL at the beginning of the experiment) (Figure 2E). Thus, the pLK and imipenem combinative effect is greater than the sum of their respective separate activities and the combination has a fractional inhibitory concentration (FIC) index equal to 0.45 [25,26]. In consequence, the pLK and imipenem combination is synergistic against this imipenem-sensitive clinical strain.

### 2.3. In Vitro and Ex Vivo Synergistic Effect of the Association of pLK and Imipenem against PAO1ΔoprD and Imipenem-Resistant Clinical P. aeruginosa Strains

PAO1*ΔoprD* strain presents an OprD porin modification, making it resistant to imipenem. For this mutant, MICs were 5 µM for pLK (same as “wild-type” PAO1) and as expected >16 µg/mL for imipenem. Our results showed a synergistic effect for the association of 2 µM pLK and 4 µg/mL imipenem (Figure 3A–C) as well as a bactericidal effect resulting in a reduction in bacterial growth of 7-log_10_ in comparison to the control (1 × 10^7^ CFU/mL at the beginning of the experiment; Figure 3C).

Then, we tested these combinations on a clinical isolate of imipenem-resistant *P. aeruginosa.* We found a synergistic effect with 1 µM pLK and 8 µg/mL imipenem and a bactericidal effect resulting in reduced bacterial growth of 5-log_10_ compared to the control (1 × 10^5^ CFU/mL at the beginning of the experiment) (Figure 3D,E). These results revealed that a combination of pLK with imipenem could contribute to counteracting *P. aeruginosa* exhibiting porin resistance.

Next, we evaluated the ex vivo effect of the pLK/imipenem combination on the PAO1*ΔOprD* strain using human primary bronchial epithelial cells (PBEC) maintained in an air–liquid interface (ALI). This 3D cell culture model partially reconstitutes the environment of human bronchial epithelium. Cells were infected by PAO1*ΔOprD* at an MOI = 0.01. One hour later, the following treatments were applied: PBS, 2 μM pLK, and 2 μg/mL imipenem alone or in combination with pLK at the same concentrations. Of note is that we previously verified the absence of the cytotoxic effect of 2 μM pLK on human bronchial epithelial BEAS-2B cells [23]. Twenty-four hours postinfection, the bacterial count was realized. The results revealed a synergistic effect for the combination of 2 µM pLK and 2 µg/mL imipenem (Figure 4), with a 3-log_10_ reduction in bacterial growth compared to the control (1 × 10^5^ CFU/mL at the beginning of the experiment).

### 2.4. In Vitro Synergistic Effect of the Association of pLK and Ceftazidime or Aztreonam on MexAB-OprM and MexXY/OprM Clinical P. aeruginosa Isolates

A MexAB-OprM isolate presents an overproduction of its efflux pump. For such multidrug-resistant *P. aeruginosa*, MICs were 2 µM for pLK, 4 µg/mL for ceftazidime and >8 µg/mL for aztreonam. We found a synergistic effect of the association between 1 µM pLK and 1 µg/mL ceftazidime, and a reduction in bacterial growth of 2-log_10_ in comparison to the control (1 × 10^5^ CFU/mL at the beginning of the experiment; Figure 5A,B). The combination of pLK and aztreonam also showed a synergistic effect with 1 µM pLK and 4 µg/mL aztreonam, and a reduction in bacterial growth of 2-log_10_ in comparison to the control (1 × 10^5^ CFU/mL at the beginning of the experiment; Figure 5C,D).

A MexXY/OprM clinical strain presents an overproduction of its efflux pump. MICs were determined at 2 µM for pLK, and at 4 µg/mL for ceftazidime. Our results showed that the combined effect of 1 µM of pLK with 1 µg/mL of ceftazidime was greater than the additive effect of each molecule individually. The association of pLK with ceftazidime also revealed a reduction in bacterial growth of 4-log_10_ or 5-log_10_, respectively, in comparison to the control (1 × 10^5^ CFU/mL at the beginning of the experiment) (Figure 6A,B).

Next, we evaluated the ex vivo effect of the pLK/ceftazidime combination on MexAB-OprM overexpressing *P. aeruginosa* strains, using human PBEC maintained in ALI. Cells were infected by the MexAB-OprM strain at an MOI = 0.01. One hour later, the following treatments were applied: PBS, 2 μM pLK, 1 μg/mL ceftazidime alone, or in combination with pLK at the same concentrations. Twenty-four hours postinfection, the bacterial count was performed. Results revealed a synergistic effect for the combination of 2 µM pLK and 1 µg/mL ceftazidime (Figure 7), with a reduction in bacterial growth of 3-log_10_ in comparison to the control (1 × 10^5^ CFU/mL at the beginning of the experiment). Hence, the combination of pLK with ceftazidime or aztreonam could hinder the efflux pump-driven resistance of *P. aeruginosa.*

## 3. Discussion

In the past decades, the overuse of antibiotics has led to the emergence of *P. aeruginosa*-resistant strains, especially β-lactams, carbapenems, and aminoglycosides, which used to be the first line of defense against this opportunistic pathogen [27]. The underlying resistance mechanisms include the enzymatic modification of antibiotics, the activation of drug efflux pumps, changes in outer membrane permeability by the negative regulation of OprD porin as well as gene mutation. Despite the clear need for new antibiotics, such drugs are slow in coming; the last entirely original antibiotic was discovered in the late 1980s [28]. Consequently, approaches that can enhance and rescue current antibiotic action are of great interest and have been increasingly studied [29,30]. A promising strategy to restore antibiotic effectiveness is the use of adjuvant molecules (also called “sensitizers”) in combination with an antibiotic [30,31].

Our previous studies showed that the cationic polypeptide pLK possesses multiple protective properties, including mucolytic activity by compacting DNA as well as antibacterial and antibiofilm activities against *P. aeruginosa* [23,24]. pLK (α-poly-L-lysine) is an organic polymer composed of lysine. There are two enantiomers of poly-lysine: L-lysine and D-lysine, each comprising two forms of poly-lysine: α-poly-lysine and ε-poly-lysine. Another form of poly-L-lysine, i.e., ε-poly-L-lysine, has been used for food preservation in several countries and is already known as an antimicrobial compound effective against *P. aeruginosa* [32,33,34,35]. Regarding the compaction properties of pLK in CF lung secretions, this represents a possible alternative for liquefying secretions, improving mucociliary clearance, and favoring the control of lung-degrading proteases by exogenous inhibitors [33,36].

In the present study, we demonstrate that pLK drastically alters *P. aeruginosa* morphology and membrane integrity, as evidenced by the vesicles visible at the bacterial surface, using electron microscopy. It is noteworthy that those pLK-triggered cell wall protuberances are similar to those induced by a distinct synthetic cationic antimicrobial peptide [27]. The exact molecular mechanism for such a pLK effect has not been established. However, we found that pLK depolarizes and permeabilizes bacterial membranes of *P. aeruginosa*. Such permeabilization activity can have important applications in the context of antibiotic resistance. Indeed, the limited permeability of the outer membrane of *P. aeruginosa* is one of the intrinsic resistance capacities of this bacteria; it acts as a selective barrier to prevent antibiotic penetration [37]. Conversely, increased membrane permeability could improve antibiotic passage toward its target and could further enhance its efficacy. In fair agreement with this hypothesis and our current findings, previous studies showed that natural antimicrobial peptides that act through their cationic charges, also exhibit antimicrobial properties [33,38].

Our previous [23,24] and current data prompted us to evaluate the combined antibacterial activity of pLK with antibiotics commonly used for controlling *P. aeruginosa* infection. To that end, we used both antibiotic-sensitive and antibiotic-resistant (reference and clinical) strains. We first showed a synergistic, bactericidal effect of the pLK–imipenem association. We further used the PAO1*ΔOprD* strain and imipenem-resistant clinical isolates to demonstrate that pLK antagonizes porin-driven drug resistance. Using MexAB-OprM and MexXY/OprM clinical *P. aeruginosa* isolates (which overproduce efflux pumps), we further showed that pLK restores ceftazidime and aztreonam activities even in drug-resistant isolates. Remarkably, the activity concentrations of pLK do not impair host cells. This is consistent with previous studies which reported that pLK is relatively nontoxic for mammalian cells because it interacts more readily with negatively charged headgroups [39]. Moreover, we previously confirmed, in vivo, the safety of the molecule, using a mouse model [23]. Here, we validated the synergistic effect of pLK and imipenem or ceftazidime in an ex vivo model of human PBEC cultured in ALI.

Hence, pLK could be a new option for treating multidrug-resistant *P. aeruginosa* infections [40]. Our results reinforce the concept that components capable of disrupting bacterial cell membranes are relevant options for fighting against antibiotic resistance. In that respect, our study supports recent data showing that a polyaminoisoprenyl compound and a synthetic peptide that binds to bacterial lipopolysaccharides [41,42,43] re-sensitizes *P. aeruginosa* to antibiotics by increasing antibiotic accumulation inside the bacteria. Of note, pLK could be especially efficient against antibiotic resistance due to bacterial efflux pump production/overexpression, porins mutations, and biofilm formation, but will not be able to cope with enzyme-dependent resistance mechanisms [43]. Regarding a limitation of our study, we have to mention its essentially observational design and limited sample size. Moreover, we will have to confirm in vivo, our current in vitro and ex vivo findings in pertinent animal models of *P. aeruginosa* infections.

Nevertheless, our findings pave the way for an innovative dual therapy against *P. aeruginosa* to limit antibiotic resistance.

## 4. Materials and Methods

### 4.1. Chemical Reagents

Poly-L-Lysine was purchased from Sigma-Aldrich (Sant-Quentin Fallavier, France) and reconstituted at 1 mM in PBS (phosphate buffered saline [137 mM NaCl, 2.7 mM KCl, 8.1 mM Na_2_HPO_4_, 1.5 mM KH_2_PO_4_, pH 7.2–7.4, 0.2 μm filtered]. Ceftazidime, imipenem/cilastatin, and gentamicin sulfate salt were purchased from Sigma-Aldrich (Sant-Quentin Fallavier, France). 3,3′ Dipropyl thiadicarbocyanine iodide (DiSC3(5)) was obtained from Molecular Probes, Inc. (Eugene, OR, USA).

### 4.2. Bacterial Strains and Culture Conditions

For this study, six different strains of *P. aeruginosa* strains were used exhibiting natural or identified acquired mechanisms of resistance: PAO1, PAO1*ΔoprD*, imipenem-sensitive clinical strains, imipenem-resistant clinical strains, MexAB-OprM clinical strain, and MexXY/OprM clinical strain. These strains were kindly provided by Dr. Katy Jeannot (Centre National de Référence de la Résistance aux Antibiotiques, Besançon, France). Lauryl Broth (LB) medium was purchased from Sigma-Aldrich (Sant-Quentin Fallavier, France), and Mueller–Hinton (MH) medium was purchased from Biorad (Roanne, France). Tryptic Soy Agar (TSA) plates were purchased from Biomérieux (Craponne, France).

### 4.3. Scanning Electron Microscopy

For scanning electron microscopy, bacteria were fixed with 1.3% glutaraldehyde and 0.05% ruthenium red in 0.07 M cacodylate buffer, pH 7.4, postfixed in 1% (vol/vol) osmium tetroxide, dehydrated in a graded ethanol series, dried with hexamethyldisilazane, and sputter coated with platinum. The sections were examined with a Zeiss Ultra Plus scanning electron microscope.

### 4.4. Membrane Permeabilization Assay

The cytoplasmic membrane depolarization activities of pLK were determined with the membrane potential-sensitive dye diSC3(5) [44]. Briefly, overnight cultures of *P. aeruginosa* were diluted in LB medium and allowed to grow to the mid-logarithmic phase determined via growth curves produced by counting the number of cfu each hour. Bacteria were collected by centrifugation, washed three times with buffer (5 mM HEPES, pH 7.8), and resuspended in the same buffer to an optical density of 0.05 (determined at 600 nm). The outer membrane of the cells was permeabilized with 0.2 mM EDTA (pH 8.0) to enable dye uptake. Then, the cell suspension was incubated for 20 min at 37 °C under shaking (150 rpm) with 0.4 μM DiSC3(5) until dye uptake was maximal, and 100 mM KCl was added to the cell suspension to equilibrate the cytoplasmic and external K+ concentrations. Different concentrations of pLK (1, 10, and 100 µM) were then added, and the fluorescence was monitored under shaking (150 rpm) at 37 °C at an excitation wavelength of 622 nm and an emission wavelength of 670 nm after 15, 30, 60, and 90 min (TECAN Infinite 200, Lyon, France). A blank with only bacteria and the dye was used as the background. This probe is taken up by bacteria according to the magnitude of the electrical gradient of the cytoplasmic membrane and becomes concentrated in the cytoplasmic membrane, where it self-quenches its own fluorescence. Any compound that alters the permeability of the cytoplasmic membrane and thus induces depolarization will lead to the release of DiSC3(5) and a consequent increase in fluorescence.

### 4.5. Susceptibility Testing

The broth microdilution method was used to determine minimal inhibitory and bactericidal concentrations (MIC and MBC). MICs were determined in accordance with the guidelines of the Clinical and Laboratory Standards Institute [45]. Briefly, *P. aeruginosa* strains were cultured on TSA plates overnight. Three isolated colonies were suspended in 3 mL of LB medium and grown overnight at 37 °C, under agitation (200 rpm). Then, several dilutions of this fresh suspension were prepared and incubated at 37 °C for 4 h, under agitation (200 rpm), to OD_600_ of 0.3–0.6, representing the logarithmic phase. The suspension with OD_600_ between 0.3 and 0.6 was centrifuged for 10 min at 3000× *g*. Bacteria were suspended in an MH medium to obtain approximately 2 × 10^5^ CFU/mL (CFU for colony forming unit). The inoculum size was verified by plating 5-fold dilutions on TSA plates and incubating overnight at 37 °C for CFU counts.

One hundred microliter/well of the bacterial suspension was inoculated into a 96-well microtiter plate and 100 µL/well of the MH (control) or antibiotic and/or pLK was added in duplicate for each condition. The microtiter plate was incubated in a plate reader (TECAN Infinite 200, Lyon, France) for 24 h at 37 °C in ambient air. The absorbance at OD_600_ was read at 30 min intervals. After incubation, the entire volume (100 µL) of each well was spread across the center of a blood agar plate, and a sterile spreading rod was used to evenly disperse the inoculum over the entire surface of the plate, which was then incubated at 37 °C for 24 h. The MBC was recorded as the lowest dilution that produced a reduction in growth ≥99.99% (≥4-log_10_ reduction in CFU/mL) compared to the control growth.

### 4.6. Synergy Determination

The synergic effects of the pLK and antibiotic combinations were defined concordantly with the usual definition of “synergy” in microbiology: “*when the effect observed with a combination is greater than the sum of the effects observed with the two drugs independently*” [46]. Moreover, the value of the fractional inhibitory concentration index (FIC index) was also used as a predictor of synergy between pLK and antibiotics. The FIC index was calculated using the following equation: ΣFIC = FIC_A_ + FIC_B_ = (C_A_/MIC_A_) + (C_B_/MIC_B_), where MIC_A_ and MIC_B_ are the MICs of molecules A and B alone, respectively, and C_A_ and C_B_ are the concentrations of A and B used in combination, respectively [26].

### 4.7. Air–Liquid Interface (ALI) PBEC

Human primary bronchial epithelial cells (PBEC, Passage #1) were thawed at 37 °C in a water bath and amplified in a 75 cm^2^ flask in KSFM Medium with human recombinant EGF at 2.5 µg/mL, BPE at 25 µg/mL, and antibiotics (penicillin 10,000 U/mL and streptomycin 10 mg/mL). Then, PBEC (passage # 2) were transferred onto a porous membrane Transwell™ insert in a 24-well plate (6.5 mm diameter with 0.4 µm pore) at a density of 7 × 10^4^ cells/mL and cultured with PneumaCult-Ex medium, at 37 °C, and incubated in 5% CO_2_ for 4 to 6 days. The Transwells were coated with a mixture of 30 µg/mL PureCol, 0.75 mg/mL BSA, and 5 µg/mL human fibronectin stabilized solution. Then, PBEC cells were maintained in an air–liquid interface (ALI), using PneumaCult-ALI medium only in the basolateral chamber (500 µL/well), for 21 days. ALI-PBEC were kindly provided by Pr Pieter Hiemstra (Laboratoire de Biologie Cellulaire et d’Immunologie du Centre médical, Leiden University, Leiden, The Netherlands). PneumaCult-Ex and PneumaCult-ALI media were purchased from Stemcell (Grenoble, France). Penicillin–streptomycin solution was purchased from Pan BioTech GmbH (Aidenbach, Germany). Phosphate buffered saline (PBS), keratinocyte serum-free medium (KSFM), human recombinant epithelial growth factor (EGF), bovine pituitary extract (BPE), bovine serum albumin (BSA) and trypsin–EDTA were purchased from Gibco (Grenoble, France). Transwell^®^ culture inserts (6.5 mm diameter, 0.4 μM pore size) were purchased from Corning Costar (Thermo Fisher, Illkirch, France). PureCol was purchased from Advanced BioMatrix (San Diego, CA, USA), and human fibronectin stabilized solution was purchased from Clinisciences (Nanterre, France).

Differentiated ALI-PBEC were infected with PAO1*ΔoprD* (in the logarithmic phase) at a multiplicity of infection (MOI) of 0.01 in the apical chamber. At one-hour postinfection, the cells were washed once from the apical side using warm PBS, followed by the addition of 100 µL of PBS or pLK with/or imipenem. Then, cells were incubated at 37 °C for 20 h in a 5% CO_2_ incubator. After incubation, ice-cold water (300 µL) was used to remove cells and bacteria from the inserts. One hundred microliters of each insert were spread on a blood agar plate and incubated at 37 °C for 24 h. The MBC was recorded as the lowest dilution that produced a reduction in growth ≥ 99.99% (≥4-log_10_ reduction in CFU/mL) compared to the control growth.

### 4.8. Statistical Analysis

Statistical analyses were performed using GraphPad Prism version 8 for Windows (GraphPad Software, www.graphpad.com). Data are reported as mean ± SEM. Statistical values, including the number of replicates (n) and the statistical test used, can be found in the figure legends. * *p* < 0.05, ** *p* < 0.005, *** *p* < 0.0005, and **** *p* < 0.0001.

## 5. Conclusions

Strains of *Pseudomonas aeruginosa* are known to possess high levels of intrinsic and acquired resistance mechanisms to counter most antibiotics, such as porin modification and efflux pump overproduction (Pang et al., 2019). In this study, we evaluated the efficacy of poly-L-lysine (pLK) in combination with commonly used antibiotics as an alternative against *P. aeruginosa*. 

First, we demonstrated that pLK could permeabilize the bacteria membrane, leading to potential use in combination with antibiotics.

Second, we demonstrated the synergistic effect of the pLK and imipenem combination against imipenem-resistant strains, using an in vitro assay. Then, this effect was also validated in a physiological model using ALI-PBEC. These results revealed that a combination of pLK with imipenem could contribute to counteracting *P. aeruginosa* which was exhibiting porin resistance.

Third, we demonstrated the synergistic effect of pLK in combination with ceftazidime or aztreonam against ceftazidime or aztreonam-resistant strains of *P. aeruginosa* which overproduce efflux systems pumps. Moreover, we confirmed this synergistic effect using ALI-PBEC. Altogether, these data revealed that pLK, a polycationic peptide, has a synergistic effect in combination with a broad range of antibiotics against sensitive, resistant, or clinical strains, with in vitro and ex vivo models.

The wide range of properties of cationic polypeptides, such as pLK or its derivatives, seem to be a promising alternative to synergize with already used therapeutic antibiotics. Hence, this offers a new opportunity to propose a dual therapy against *P. aeruginosa* to limit the emergence of antibiotic resistance.

## Figures and Tables

**Figure 1 ijms-24-02851-f001:**
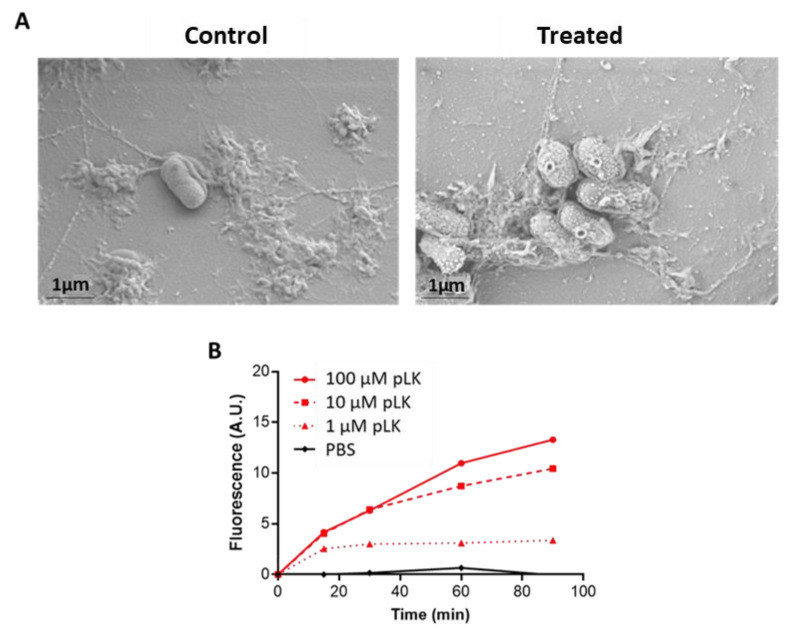
Effect of pLK on the morphology and permeability of the *P. aeruginosa PAO1* strain. (**A**) Scanning electron micrographs of *P. aeruginosa* exposed or not to 10 μM pLK. (**B**) Effect of pLK on *P. aeruginosa* membrane polarization. Results are expressed in relative intensity fluorescence observed at 670 nm compared to the negative control (PBS). Values are mean ± SEM of three independent determinations.

**Figure 2 ijms-24-02851-f002:**
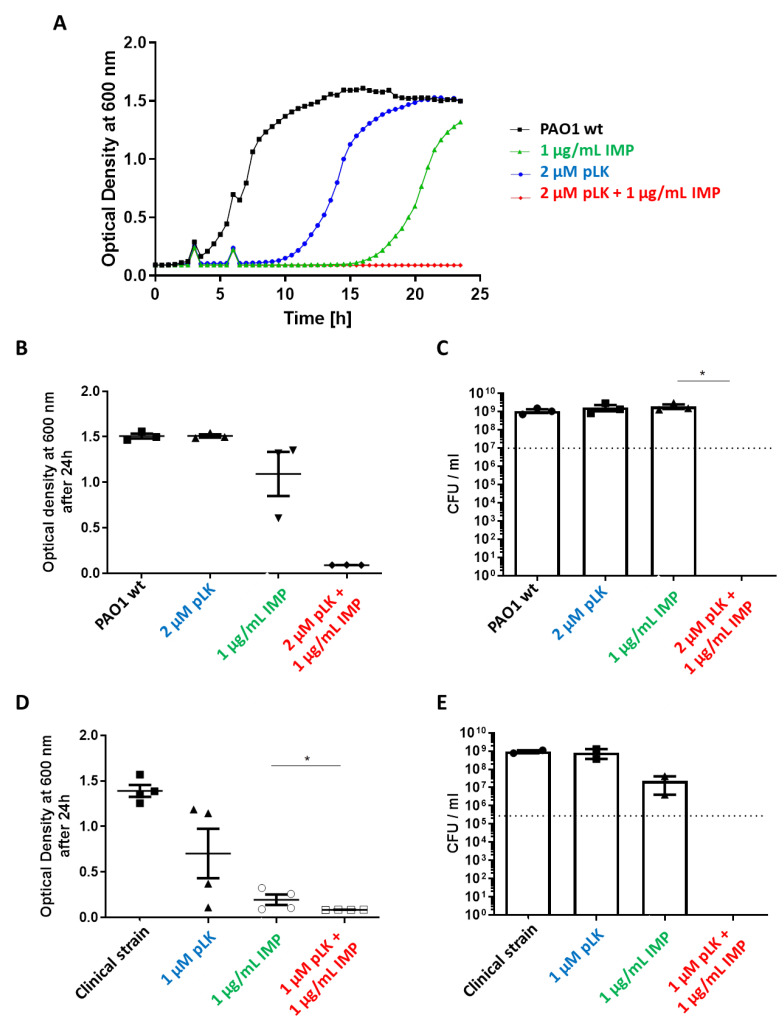
In vitro effect of the pLK and imipenem association against the reference (PAO1) and imipenem-sensitive *P. aeruginosa* strains. PAO1 (1 × 10^7^/mL in exponential phase) and an imipenem-sensitive clinical *P. aeruginosa* (1 × 10^5^/mL in exponential phase) strain were incubated at 37 °C in Mueller–Hinton medium in the presence or absence of 2 uM (**A**–**C**) or 1 uM (**D**,**E**) poly-L-Lysine (pLK), 1 µg/mL imipenem (IMP) or their combination. (**A**,**B**,**D**) Bacterial growth was measured by monitoring the optical density at 600 nm after 24 h of incubation. (**C**,**E**) Determination of the colony forming unit (CFU)/mL was evaluated after 24 h of incubation. Values are mean ± SEM of three independent experiments. Statistical analysis was performed using the Mann–Whitney test. * *p* < 0.05.

**Figure 3 ijms-24-02851-f003:**
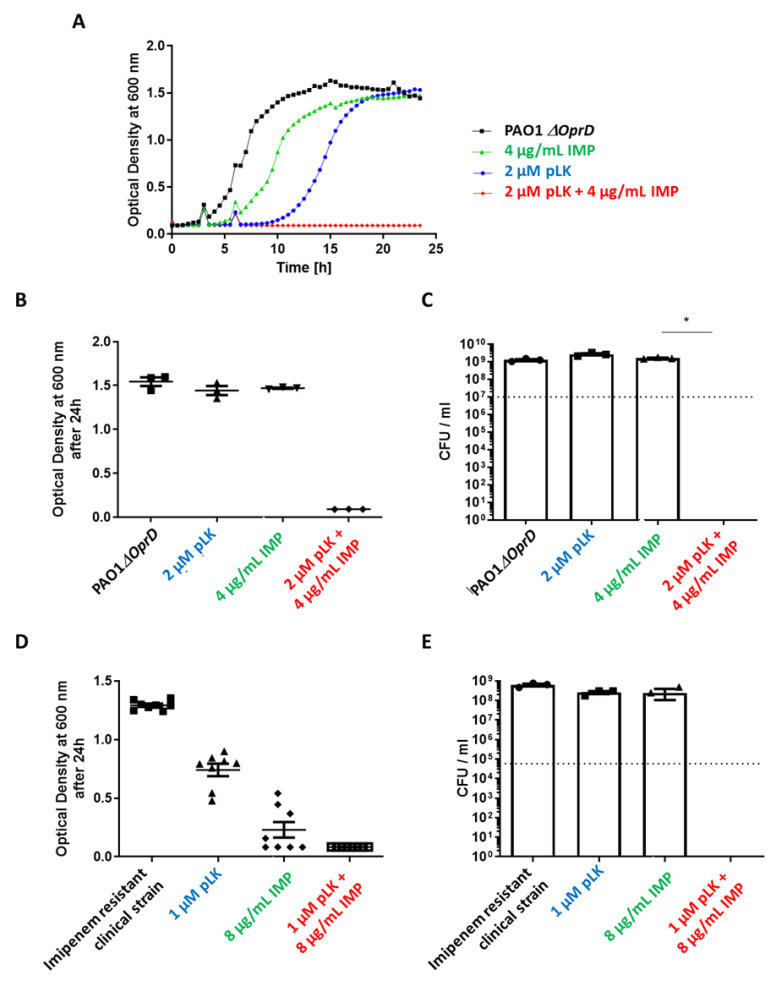
In vitro synergistic effect of the association between pLK and imipenem against PAO1*ΔoprD* and imipenem-resistant clinical *P. aeruginosa* strains. PAO1*ΔoprD* strain (1 × 10^7^ /mL in exponential phase) and an imipenem-resistant isolate (1 × 10^5^ /mL in exponential phase) were incubated at 37 °C in Mueller–Hinton medium in the presence or absence of 2 µM poly-L-Lysine (pLK), 4 or 8 µg/mL imipenem (IMP) or their combination. (**A**,**B**,**D**) Bacterial growth was measured by monitoring the optical density at 600 nm after 24 h of incubation. (**C**,**E**) Determination of the colony forming unit (CFU)/mL was evaluated after 24 h of incubation. Values are mean ± SEM of three independent experiments. Statistical analysis was performed using the Mann–Whitney test. * *p* < 0.05.

**Figure 4 ijms-24-02851-f004:**
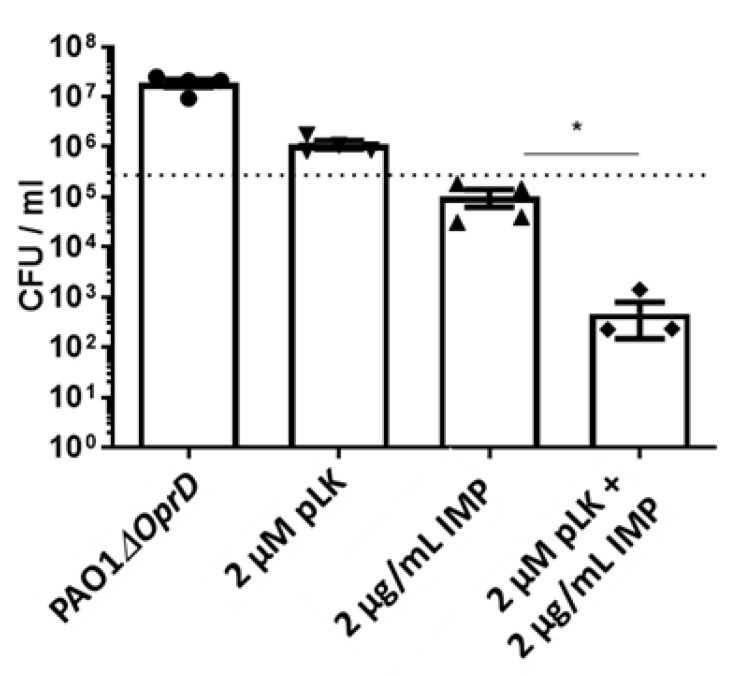
Ex vivo synergistic effect of the association between pLK and imipenem against an imipenem-resistant *P. aeruginosa* strain. Human primary bronchial epithelial cells maintained in an air–liquid interface were infected with PAO1*ΔOprD* strain and treated one hour postinfection with either PBS, 2 μM pLK, 2 μg/mL imipenem or with a combination of the two molecules. Determination of the colony forming unit (CFU)/mL was evaluated 24 h postinfection. Values are mean ± SEM of three independent experiments. Statistical analysis was performed using the Mann–Whitney test (* *p* < 0.05).

**Figure 5 ijms-24-02851-f005:**
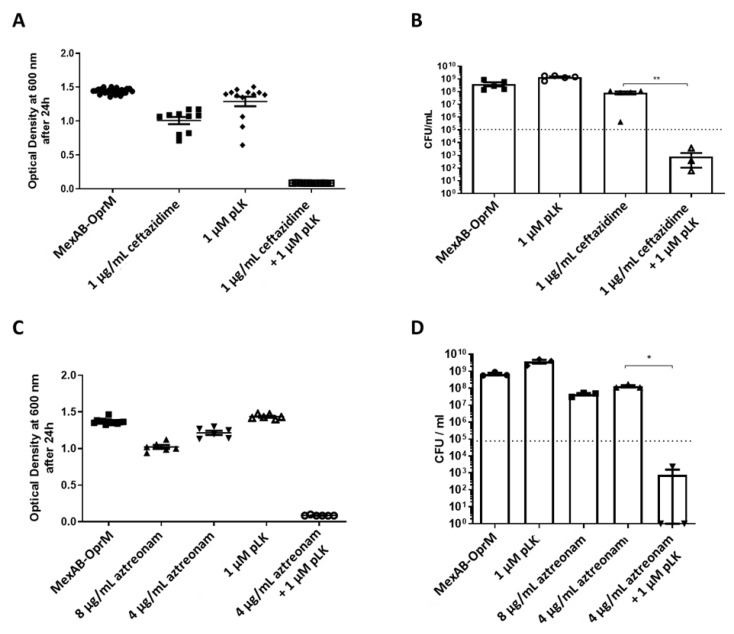
In vitro synergistic effect of pLK/ceftazidime or pLK/aztreonam combinations on a *P. aeruginosa* strain overproducing MexAB-OprM efflux pump. (**A**,**C**) Bacteria (1 × 10^5^ /mL in exponential phase) were incubated at 37 °C in Mueller–Hinton medium in the presence or absence of 1 µM pLK, 1 µg/mL ceftazidime, 4 or 8 µg/mL of aztreonam, or the combination of the antibiotic with pLK. Bacterial growth was measured by monitoring the optical density at 600 nm after 24 h of incubation. (**B**,**D**) Determination of the colony forming unit (CFU)/mL was evaluated after 24 h of incubation. Values are mean ± SEM of five independent experiments. Statistical analysis was performed using the Mann–Whitney test (* *p* < 0.05, ** *p* < 0.005).

**Figure 6 ijms-24-02851-f006:**
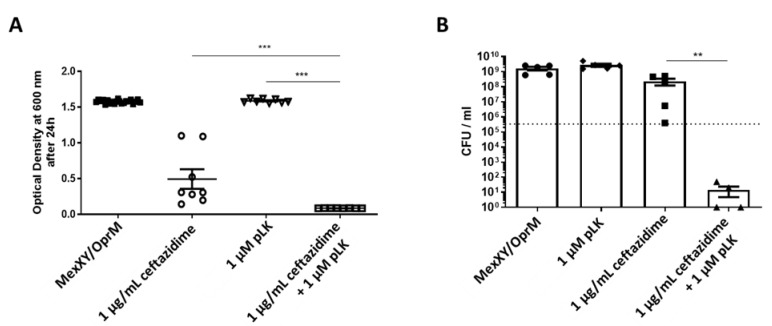
In vitro synergistic effect of the combination of pLK and ceftazidime on a *P. aeruginosa* strain overproducing MexXY/OprM. (**A**) Bacteria (1 × 10^5^ /mL in exponential phase) were incubated at 37 °C in Mueller–Hinton medium in the presence or absence of 1 µM pLK, 1 µg/mL ceftazidime, or their combination. Bacterial growth was measured by monitoring the optical density at 600 nm after 24 h of incubation. (**B**) Determination of the colony forming unit (CFU)/mL was evaluated after 24 h of incubation. Values are mean ± SEM of five independent experiments. Statistical analysis was performed using the Mann–Whitney test (** *p* < 0.005, *** *p* < 0.0005).

**Figure 7 ijms-24-02851-f007:**
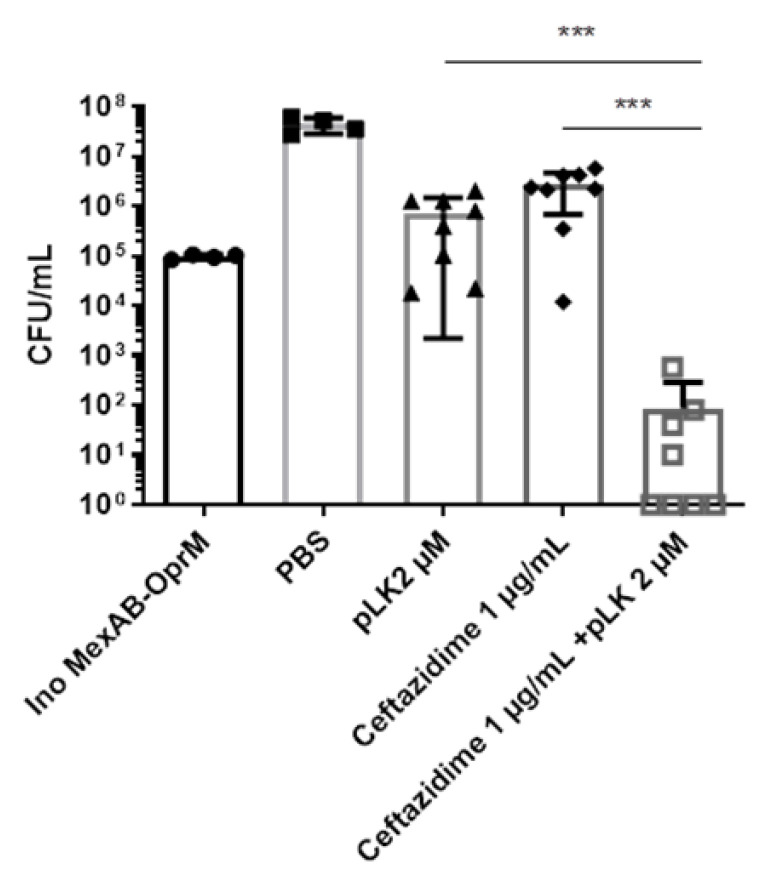
Ex vivo synergistic effect of the combination of pLK and ceftazidime against a ceftazidime-resistant *P. aeruginosa* strain. Human primary bronchial epithelial cells maintained in an air–liquid interface were infected with a MexAB-OprM clinical strain and treated one hour postinfection with either PBS, 2 μM pLK, 1 μg/mL ceftazidime or the combination of ceftazidime/pLK. Determination of the colony forming unit (CFU)/mL was evaluated after 24 h postinfection. Values are mean ± SEM of three independent experiments. Statistical analysis was performed using the Mann–Whitney test (*** *p* < 0.0005).

## Data Availability

Data sharing is not applicable.

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
