# Peer review of "Poly-L-Lysine to Fight Antibiotic Resistances of Pseudomonas aeruginosa"

_ijms, 2023, doi:10.3390/ijms24032851_

Round 1
Reviewer 1 Report
The manuscript entitled "Poly-L-lysine to fight antibiotic resistances of Pseudomonas aeruginosa" delivered a detailed overview and excellent information on how pLK alters bacterial membrane integrity and permeabilizes the bacterial membrane of P. aeruginosa using various approaches. The authors have significantly contributed to describing the possible eradication of bacterial infection using the combination of pLK and antibiotics. This is a detailed, concise, authentic, and well-written manuscript. The introduction is relevant and statics based. Sufficient information about the previous study findings is presented for readers to follow the present study rationale. Though the manuscript is written well, some modifications in a few sentences and adding details and references could improve it. Given these shortcomings, the manuscript requires minor revisions, and I believe the manuscript is suitable for publication after the authors have addressed the following comments and questions.
1. Abstract
Comment: In the abstract, please add a few lines for applying the combination of pLK and antibiotics strategies together as a take-home message, and it should be better explained and emphasized with anti-biotherapy.
2. Introduction- Para 1.1.-Current trends of inappropriate.... 10 million deaths worldwide by year in 2050.
Comment: Add the reference for these statics. Are they recent statistics?
2-Authors: Line 78.
Comments: Authors must use the full abbreviation of CF here as they have introduced this first time.
3-Authors: Lines 88-90
Comments: Please provide appropriate references here.
4-Authors: Results- Lines 103-108
Comments: Please explain this section appropriately regarding your results. In other words, please describe the significance of your results in one or two lines.
5-Authors: Lines 182-188, section 2.3
Comment: In this paragraph, the authors have discussed the synergistic effect of the association of pLK and imipenem against PAO1doprD and imipenem-resistant clinical P. aeruginosa strains. However, the authors must have described these results reasonably, such as what you observed here and what is the essence of these results.
5-Authors: Discussion section, Lines 331-337
Comment: Can you clarify what the derivatives of pLK could be used as a promising option? Are there any other studies available on the same? If yes, please describe those studies in line with your outcomes in this section.
Author Response
Thank you for your reply and your comments.
Please find our answers for each point below :
Point 1: Abstract
Comment: In the abstract, please add a few lines for applying the combination of pLK and antibiotics strategies together as a take-home message, and it should be better explained and emphasized with anti-biotherapy.
Response 1 : Abstract was completed according to your suggestions : “Pseudomonas aeruginosa is a major hospital-associated pathogen that can cause severe infections, most notably in patients with cystic fibrosis (CF) or those hospitalized in intensive care units. Given its remarkable ability to resist antibiotics, P. aeruginosa eradication has grown more chal-lenging. Therefore, discovery and development of new strategies that can counteract P. aerugino-sa resistant strains become urgent.
Here, we evaluated the efficacy of poly-L-lysine (pLK) in combination with commonly used an-tibiotics, as an alternative treatment option against P. aeruginosa. First, we demonstrated by scanning electron microscopy that pLK alters the integrity of the surface membrane of P. aeru-ginosa. We also showed using a fluorometry test that this results in an enhanced permeability of the bacteria membrane. Based on these data, we further evaluated in vitro by broth microdilution method the effect of combination of pLK with imipenem, ceftazidime or aztreonam. We found synergies in terms of bactericidal effect against either sensitive or resistant P. aeruginosa strains, with a reduction of bacterial growth (up to 5-log10 in comparison to the control). Similarly, these synergistic and bactericidal effects were confirmed ex vivo using a 3D model of human primary bronchial epithelial cells maintained in air-liquid interface. In conclusion, pLK could be an in-novative antipseudomonal molecule, opening its application as an antibiotherapy adjuvant against drug-resistant P. aeruginosa strains.”
Point 2: Introduction- Para 1.1.-Current trends of inappropriate.... 10 million deaths worldwide by year in 2050.
Comment: Add the reference for these statics. Are they recent statistics?
Response 2: Reference from 2016 has been added : “O’Neill J Tackling Drug-Resistant Infections Globally: Final Report and Recommendations. Review on Antimicrobial Resistance London2016.”
Point 3: Line 78.
Comments: Authors must use the full abbreviation of CF here as they have introduced this first time.
Response 3: Full abbreviation has been added.
Point 4: Authors: Lines 88-90
Comments: Please provide appropriate references here.
Response 4: Reference has been added : Smith, W.D.; Bardin, E.; Cameron, L.; Edmondson, C.L.; Farrant, K.V.; Martin, I.; Murphy, R.A.; Soren, O.; Turnbull, A.R.; Wierre-Gore, N.; et al. Current and Future Therapies for Pseudomonas Aeruginosa Infection in Patients with Cystic Fibrosis. FEMS Microbiology Letters 2017, 364, doi:10.1093/femsle/fnx121.
Point 5: Results- Lines 103-108
Comments: Please explain this section appropriately regarding your results. In other words, please describe the significance of your results in one or two lines.
Response 5: We completed this section as follows : “P. aeruginosa morphology was examined by scanning electron microscopy to visu-alize the effect of pLK on bacteria (Fig. 1A). The bacterial surface was uniform and smooth without pLK (Fig. 1A, left panel), but following an incubation with 10 μM pLK, alterations occurred at the bacterial surface. (Fig. 1A, right panel). This result may suggest permeabilization of the bacterial membrane.”
Point 6: Lines 182-188, section 2.3
Comment: In this paragraph, the authors have discussed the synergistic effect of the association of pLK and imipenem against PAO1doprD and imipenem-resistant clinical P. aeruginosa strains. However, the authors must have described these results reasonably, such as what you observed here and what is the essence of these results.
Response 6: Supplements have been added to this part : “These results revealed that combination of pLK with imipenem could contribute to counteract P. aeruginosa exhibiting porin resistance.”
Point 7: Discussion section, Lines 331-337
Comment: Can you clarify what the derivatives of pLK could be used as a promising option? Are there any other studies available on the same? If yes, please describe those studies in line with your outcomes in this section.
Response 7: Thank you to the reviewer for this important issue. Accordingly, we revised the discussion as follows (see also lanes 334-344).
“Our previous studies showed that the cationic polypeptide pLK possesses multiple protective properties, including a mucolytic activity by compacting DNA as well as anti-bacterial an anti-biofilm activities against P. aeruginosa [23,24]. pLK (α-poly-L-lysine) is an organic polymer composed of lysine. There are two enantiomers of poly-lysine: L-lysine and D-lysine, each comprising two forms of poly-lysine: α-poly-lysine and ε-poly-lysine. Another form of poly-L-lysine, i.e. ε-poly-L-lysine, has been used for food preservation in several countries, and is already known as an antimicrobial compound effective against P. aeruginosa [32,34–36]. Regarding the compaction properties of pLK in CF lung secretions, this represents a possible alternative for liquefying secretions, improving mucociliary clearance, and favoring the control of lung-degrading proteases by exogenous inhibitors [32,33]”.

Reviewer 2 Report
Find my comments in the attached file. Use Adobe Reader to see them

Author Response
Thank you for your reply and your comments.
Please see the attached file for the requested modifications (highlighted in yellow).
For your other questions, please find our answers below:
Figure 1. Move close to the first citation (Line 109).
The scalebare in panel A reports 1 μM instead of 1 μm. Moreover, change "without pLK" and "10 μM pLK" with "Control Group (or CTRL group)" and "Treated Group (or TRT Group)"
We placed this figure after the paragraph as requested in the journal guidelines.
Lines 142-143: it is not clear why you have used these concentrations (pLK 2 μM and imipenem 1 mg/L).
“Several concentrations of pLK and imipenem were tested, the concentration of each com-ponent showing no bacteriostatic effect was retained.”
Lines 145,149: in method section you did not describe how to determine the synergy when combining pLK and antibiotics. I strongly recommend to use FIC index or other index(es).
We have added a section synergy determination in the Materials and Methods part :“4.6. Synergy determination. Synergic effect of pLK and antibiotic combination was define concordantly with the usual definition of “synergy” in microbiology : when the effect ob-served with a combination is greater than the sum of the effects observed with the two drugs independently (Acar, 2000). The value of the FIC index was also used as a predictor of syn-ergy between pLK and antibiotic (Hall et al., 1983)”
Line 346: clearly state the number of strains included and tested in your work. For this study, 6 different strains of P. aeruginosa strains were used exhibiting natural or identified acquired mechanism of resistance: PAO1, PAO1oprD, imipenem-sensitive clinical strains, imipenem-resistant clinical strains, MexAB-OprM clinical strain and MexXY/OprM clinical strain.
Line 374: which are the volts used?
3,3′-Dipropylthiadicarbocyanine (diSC3(5)) is a fluorescent dye that has been used to monitor cell membrane potential.1,2,3 It displays excitation/emission maxima of 622/670 nm, respectively, and, upon cell hyperpolarization, it enters cells, and exhibits a shift in emission maxima to 688 nm and a decrease in fluorescence intensity.1 When the cell membrane is depolarized, the fluorescence intensity of diSC3(5) increases as it exits the cells.
400-406: why have you used this method to reach 10^5? Normally, fresh overnight cultures are used, and 3-4 well separeted colonies are dispersed in PBS to reach 10^8 CFU/mL and then diluted to 10^5.
The protocol we used is similar to yours. The suspension obtained has an OD600 of 0.3–0.6, and contains about 10^8. It is further centrifuged and diluted.

Reviewer 3 Report
The manuscript by Cezard et al. has scientific value, is well designed and conducted, with original and relevant contributions in the fight against antibiotic resistance of Pseudomonas aeruginosa. The use of antimicrobial compounds with synergistic or potentiating activity of some antibiotics appears to be an effective alternative to combat the phenomenon of antimicrobial resistance, with a global spread.
I support its publication after appropriate minor modifications, as outlined below.
In the Materials and methods section, the subchapters are not numbered. Please number them.
Please insert the references for the methods used in the research.
I strongly recommend the introduction of a 'Conclusions' chapter, focusing on a succinct presentation of the conclusions derived from the obtained results and highlighting the practical applicability of the study results. Limitations of the study and future perspectives should also be addressed.
Please respect the requirements of the journal as regards the preparation of the list of references.
The name of the microorganism species is not italicized (lines 485, 490, 501, 504, 507, 512, 515, 517, 522, 524, 527). Please be carefully with this basic concern throughout the manuscript!
Author Response
Thank you for your reply and your comments.
Please find our answers for each point below :
Point 1: In the Materials and methods section, the subchapters are not numbered. Please number them.
Response 1: Materials and methods section has been modified according to your suggestions.
Point 2: Please insert the references for the methods used in the research.
Response 2: Materials and methods section has been modified according to your suggestions.
« Clinical and Laboratory Standards Institute. Methods for Dilution Antimicrobial Susceptibility Tests for Bacteria That Grow Aerobically, 10th Ed. Approved Standard M07-A11. 2018.”
Point 3: I strongly recommend the introduction of a 'Conclusions' chapter, focusing on a succinct presentation of the conclusions derived from the obtained results and highlighting the practical applicability of the study results. Limitations of the study and future perspectives should also be addressed.
Response 3: A conclusion part has been separated upstream from the discussion part. Applicability and limitations were discussed in the discussion section.
Point 4: Please respect the requirements of the journal as regards the preparation of the list of references.
Response 4: list of references was modified in accordance with journal requirements.
Point 5: The name of the microorganism species is not italicized (lines 485, 490, 501, 504, 507, 512, 515, 517, 522, 524, 527). Please be carefully with this basic concern throughout the manuscript!
Response 5: The name of the microorganism species has been italicized everywhere in the manuscript.
Round 2
Reviewer 2 Report
Find minor revisions in the attached file. Use Adobe Reader to see them

Author Response
Dear Reviewer 2,
Thank you for all your corrections. Please find the attached file with all requested modifications highlighted in yellow.
Respectfully,
Virginie HERVE.
